# Semantic Segmentation of Hyperspectral Remote Sensing Images Based on PSE-UNet Model

**DOI:** 10.3390/s22249678

**Published:** 2022-12-10

**Authors:** Jiaju Li, Hefeng Wang, Anbing Zhang, Yuliang Liu

**Affiliations:** 1School of Mining and Geomatics Engineering, Hebei University of Engineering, Handan 056038, China; 2Key Laboratory of Natural Resources and Spatial Information, Handan 056038, China; 3Hydrogeology Team of Hebei Coalfield Geology Bureau, Handan 056038, China

**Keywords:** hyperspectral remote sensing images, dataset partition method, convolutional neural networks, PSE-UNet, Hughes phenomenon, semantic segmentation

## Abstract

With the development of deep learning, the use of convolutional neural networks (CNN) to improve the land cover classification accuracy of hyperspectral remote sensing images (HSRSI) has become a research hotspot. In HSRSI semantics segmentation, the traditional dataset partition method may cause information leakage, which poses challenges for a fair comparison between models. The performance of the model based on “convolutional-pooling-fully connected” structure is limited by small sample sizes and high dimensions of HSRSI. Moreover, most current studies did not involve how to choose the number of principal components with the application of the principal component analysis (PCA) to reduce dimensionality. To overcome the above challenges, firstly, the non-overlapping sliding window strategy combined with the judgment mechanism is introduced, used to split the hyperspectral dataset. Then, a PSE-UNet model for HSRSI semantic segmentation is designed by combining PCA, the attention mechanism, and UNet, and the factors affecting the performance of PSE-UNet are analyzed. Finally, the cumulative variance contribution rate (CVCR) is introduced as a dimensionality reduction metric of PCA to study the Hughes phenomenon. The experimental results with the Salinas dataset show that the PSE-UNet is superior to other semantic segmentation algorithms and the results can provide a reference for HSRSI semantic segmentation.

## 1. Introduction

Hyperspectral imaging technology can simultaneously obtain 3D spatial and spectral information of land features. Thus, it has a prominent advantage in the fine-grained land cover classification of remote sensing images and has been widely used in agriculture, forestry, military, mineral recognition, and marine research [1,2,3,4,5]. The sematic segmentation of hyperspectral remote sensing images (HSRSI) faces several technical challenges such as a complex data structure, massive computation, and high information redundancy [6,7]. The traditional machine learning classification method that needs to manually design features can no longer meet the needs of hyperspectral data [8]. Therefore, there is an urgent need for an efficient and intelligent classification technique for HSRSI.

With the rapid development of deep learning technology, the algorithm of convolutional neural networks (CNN) has been widely used in many fields, including image classification, sematic segmentation, and video understanding [9,10,11,12,13,14], and has become a research hotspot in the land cover classification of HSRSI. In 2015, Hu et al. constructed a 1D-CNN model of “convolutional-pooling-fully connected” structure to extract the spectral information of HSRSI and obtained higher classification accuracy than the support vector machine (SVM) and deep neural networks (DNN) [15,16,17]. However, due to the phenomena of “different objects with the same spectrum” and “different spectra for the same object”, only extracting spectral information limits the performance of the CNN classifier. At the same time, the method used in the field of computer vision for extracting the spatial features of images has been used in several studies to extract the spatial information of HSRSI by constructing a 2D-CNN based on 2D convolution [18]. However, the “dimension disaster” caused by the small sample sizes and high dimensions of HSRSI limits the performance of the 2D-CNN classifier [6]. To solve this problem, the principal component analysis (PCA) is usually used to reduce the dimension to improve the classification accuracy [19,20,21,22,23,24]. However, neither the 1D-CNN nor the 2D-CNN makes full use of the 3D information of HSRSI. Therefore, using a CNN classifier to extract the spatial and spectral joint features simultaneously has become the mainstream research direction. Currently, two methods are often used to extract the spatial and spectral joint features: one is to use the 3D-CNN based on 3D convolution to directly extract the spatial and spectral features of the hyperspectral images [25,26,27,28,29,30]; the other is to use different combinations of the 1D-CNN, 2D-CNN, and 3D-CNN to develop models for this purpose [31,32,33]. The CNN models constructed with these two methods have better performance in the classification of HSRSI than the CNN models that only extract features of a single dimension. The CNN models based on a “convolutional-pooling-fully connected” structure have made positive progress in the classification of HSRSI, but there are still issues that need to be further explored.

Firstly, to make full use of the annotation information in the hyperspectral dataset of small samples, most researchers use a sliding window with a stride of 1 to segment the images into patches and transmit them into the model. However, Nalepa et al. experimentally verified that partitioning the dataset in this way will lead to information leakage between the training set and the test set, resulting in overly optimistic classification results. Therefore, Nalepa et al. proposed a dataset partition method based on random patches. Randomly extracted multiple patches of m × n five times from the image were to be used as training data and the rest used as test data, effectively avoiding information leakage [34]. Zou et al. used the sliding window of n × n with a stride of n for non-overlapping dataset partitioning, and divided the dataset into the training set, test set, and unlabeled patches, which is simple to implement and avoids information leakage at the same time [35]. Qu et al. proposed a dataset partition method that divided the dataset into non-overlapping training, leakage, validation, and test areas. The model performance was evaluated through the training and the test areas, and the severity of information leakage was evaluated through the leakage and the test areas [36]. Although the above-mentioned studies solved the problem of information leakage, there are still some unresolved problems, such as not including all land cover classes in the training set, the lack of randomness in data distribution, and data redundancy. In addition, the labeling quality of the data is ensured by discarding the unlabeled background pixels. However, the interference of the background in practical applications cannot be avoided.

Secondly, sample sizes of HSRSI are small, and it is difficult for the CNN classification models based on a “convolutional-pooling-fully connected” structure to fully utilize the annotation information [35]. In order to improve the utilization of annotation information, Long et al. proposed fully convolutional networks (FCN) [10] based on semantic segmentation by replacing the fully connected layer in the VGG-16 [9] network with the convolution layer and using the transposed convolution to restore the image resolution, which successfully extended the classification of CNN from image-wise to pixel-wise. Zou et al. proposed the SS3FCN network and applied the FCN for the classification of the HSRSI for the first time [35]. Qu et al. proposed the TAP-Net network that used three attention mechanisms and four parallel subnetworks to enhance the extraction capacity for features of the HSRSI [36]. Although the above-mentioned models achieved good classification accuracy, due consideration has not been given to the small sample sizes and high dimensions of HSRSI in the algorithm structure. The UNet model proposed by Ronneberger et al. has achieved excellent results in the semantic segmentation of medical images that also have small sample sizes and high-resolution remote sensing images [12,37,38,39,40]. The 3D-UNet network proposed by Çiçek et al. has been successfully applied to the semantic segmentation of high-dimensional 3D medical images [41]. However, the UNet-based approaches are rarely used in the semantic segmentation of HSRSI. Moreover, the algorithm structure in UNet-based approaches still has room for improvement.

Finally, small sample sizes and high dimensions of HSRSI lead to the Hughes phenomenon [42]. Most researchers use PCA to reduce the dimensions of HSRSI to avoid the curse of dimensionality. However, there is no scientific method to define the number of principal components after dimensionality reduction. Some researchers selected three principal components by referring to RGB images [20,21], while others defined the number of principal components by experience [19,22,32]. The above-mentioned studies have all avoided overfitting caused by small sample sizes and high dimensions, but dimensionality reduction can be very subjective and cannot provide a reference for future research. Xu et al. analyzed the classification accuracy of HSRSI with its dimensionality reduced to 1 with eight principal components [24]. However, only the first few principal components are not comprehensive enough for HSRSI with hundreds of bands. Therefore, it is necessary to further analyze how the land cover classification accuracy of HSRSI changes from a low dimension to a higher dimension.

In summary, the current HSRSI semantic segmentation faces the following three challenges:Although existing dataset partition methods avoid problems of information leakage, they still suffer from two inadequacies: not including all land cover classes in the training set and discarding the unlabeled background pixels.The UNet-based approaches for sematic segmentation of HSRSI, mostly directly employing the standard UNet [43,44], are not optimized for the characteristics of the HSRSI and still have room for improvement.The PCA can overcome the impact of the curse of dimensionality on segmentation accuracy, but researchers tend to subjectively choose the number of dimensions and cannot provide a reference for future research.

In order to overcome the above challenges, firstly, this paper introduces the patch allocation scheme based on the non-overlapping sliding window strategy commonly used in computer vision into the sematic segmentation of HSRSI, and combines a judgment mechanism to make up for the disadvantage that not all classes can be included in the training set after the patches are randomly allocated. Secondly, this paper proposes a new PSE-UNet model for semantic segmentation of HSRSI. Compared with the method of directly using standard UNet [43,44], PSE-UNet considers the characteristics of HSRSI, combines UNet with PCA and the attention mechanism, reduces the performance loss caused by dimensional disasters, and enhances the expression of spectral information. In addition, considering the small number of HSRSI samples, the influence of downsampling times, different downsampling and upsampling methods, and different activation functions on segmentation performance are discussed, and the most appropriate PSE-UNet variant is determined. Finally, the cumulative variance contribution rate (CVCR) is introduced as the dimensionality reduction index to study the Hughes phenomenon and comprehensively analyze how the land cover classification accuracy of HSRSI changes from a low dimension to a higher dimension. The main contributions of this paper can be summarized as follows:The non-overlapping sliding window method combined with the judgment mechanism can effectively avoid information leakage, overcome the shortcomings of existing dataset partition methods, and provide a fair comparison between models.The proposed PSE-UNet is based on the “encoder-decoder” structure, considers the small sample sizes and high dimensions of the HSRSI, and improves the HSRSI semantic segmentation accuracy.The Hughes phenomenon in HSRSI semantic segmentation is comprehensively analyzed, which can provide a reference for determining the dimension of HSRSI dataset.

## 2. Research Methodology

The overall framework contains four steps: dataset partitioning, a training model, evaluating model performance, and predicting the segmentation map, which can be seen in Figure 1. First, the dataset is randomly divided into a training set, validation set, and testing set using the non-overlapping sliding window strategy combined with a judgment mechanism. Then, the PSE-UNet model is trained and evaluated. Finally, the best model is used to predict the segmentation map.

### 2.1. Dataset Partition Method

In order to solve the problems of existing dataset partition methods, this paper introduces the patch allocation scheme based on the non-overlapping sliding window strategy commonly used in computer vision into the sematic segmentation of HSRSI, and combines a judgment mechanism to make up for the disadvantage that not all classes can be included in the training set after the patches are randomly allocated. Considering the actual applications, the background pixels are retained and information leakage can be effectively avoided at the same time. The method can be used to fairly compare the segmentation performance of different models. The basic idea for dataset partitioning with the method is shown in Figure 2, and the specific steps are as follows:

Step 1: The non-overlapping sliding window strategy is used to cut the hyperspectral remote sensing dataset into patches of n × n in size.

Step 2: The patches are randomly assigned to the training, validation, and test sets according to the common allocation ratio of 6:2:2 for small-scale datasets.

Step 3: Repeat Step 2 until each set contains all land cover classes.

### 2.2. PSE-UNet Model

This paper proposes a PSE-UNet model based on the “encoder-decoder” structure for semantic segmentation of HSRSI, as shown in Figure 3. The PSE-UNet model is composed of a PCA module, C-SE modules, skip connections, and Softmax. The C-SE module is the basic unit of the PSE-UNet model for extracting the features. Considering the high dimensions of HSRSI, PCA and the channel attention mechanism are adopted in PSE-UNet; PCA is used to avoid the curse of dimensionality, and the channel attention mechanism is used to learn the interdependence between the feature channels. Considering the small sample sizes of HSRSI, the standard UNet can easy cause overfitting. Therefore, in PSE-UNet, the number of channels in each stage is reduced, and the appropriate downsampling times, downsampling and upsampling methods, and activation functions are selected.

The PCA is used for dimensionality reduction of the HSRSI before they are input into the encoder, and the CVCR is selected as the dimensionality reduction standard to analyze the impact of different CVCRs on the segmentation accuracy. The encoder consists of two C-SE modules and two downsampling units. The decoder consists of two C-SE modules and two upsampling units. The skip connections are used to combine the shallow and deep features to avoid the loss of spatial information caused by downsampling. After all convolution operations in the model occur, the batch normalization (BN) [45] module is connected and the parametric rectified linear unit (PReLU) [46] is used as the activation function. Finally, Softmax is used for pixel-wise classification, and the output of Softmax is the final segmentation results of land cover classes in the network.

#### 2.2.1. PCA

In PSE-UNet, we use PCA to reduce the dimensions of HSRSI to avoid the curse of dimensionality. For the input HSRSI data X, it was converted into the corresponding principal component matrix Y by PCA, and then the number of principal components to be retained is selected by the CVCR to obtain the reduced dimension data. PCA is one of the most widely used data dimensionality reduction methods, which transforms input data into linear independent variables and retains most of the information in the original data, and the specific steps are as follows:

Firstly, input data standardization is done to obtain matrix X so that the mean value of each row element is zero, and a new matrix $X^{\prime }$ is constructed using the following equation:(1)$$X^{\prime }=\frac{1}{\sqrt{n-1}}X^{T}$$

In the above equation, the mean value of each column of matrix $X^{\prime }$ is zero, and n is the sample size.

Secondly, the truncated singular value decomposition (SVD) of matrix $X^{\prime }$ is processed to obtain three matrices: U, Σ, and V. The equation is as follows:(2)$$X^{\prime }=U\Sigma V^{T}$$

Finally, the first k columns of matrix V are used to constitute the k sample principal components, and the principal component matrix Y can be obtained with the following equation:(3)$$Y=V_{k}^{T}X$$

In addition, *CVCR* is used to select the retained principal component number, which is calculated using the following equation:(4)$$CVCR=\overset{k}{\underset{i=1}{\sum }}\eta_{i}=\frac{\sum_{i=1}^{k}\lambda_{i}}{\sum_{i=1}^{m}\lambda_{i}}$$
where, $\eta_{i}$ is the variance contribution of the i-th principal component, $\lambda_{i}$ is the eigenvalue of the i-th principal component, k is the number of selected principal components, and m is the total number of principal components.

#### 2.2.2. C-SE Module

The C-SE module consists of convolutions, BN, PReLU, and an SE (Squeeze and Excitation) module [47], as shown in Figure 4. Since the HSRSI is multi-dimensional, an SE module, a lightweight channel attention mechanism, is introduced after the convolution module in the C-SE module to learn the interdependence between the feature channels through the “Squeeze-and-Excitation” structure. During the squeeze, the global average pooling (GAP) layer is used to compress the input 2D feature map into 1D real numbers, and in the excitation, two fully connected layers and a rectified linear unit (ReLU) [48] are used to build a model to fit the nonlinear relationship between the feature channels. Finally, the channel weight normalized by the sigmoid function is multiplied by the input feature map to enhance the related features and suppress the unrelated features. Hence, the C-SE module can extract more discriminative semantic features and obtain better segmentation results.

#### 2.2.3. Downsampling and Upsampling

Downsampling (or subsampling) is a way to reduce the resolution of images. The purpose of downsampling is to reduce the amount of calculation and increase the receptive field. The most commonly used downsampling method is max pooling. In this method, an operation is performed at the maximum value with the input images in a window of size n × n with a stride of n, and the maximum value of each window is taken as the pixel value of the corresponding position of the output image. In this paper, a convolution layer with a stride of 2 and a convolution kernel of size 2 × 2 is used to replace the pooling layer for downsampling the input images. The image size is reduced to half of the original after each downsampling.

Contrary to downsampling, upsampling is a way to restore image resolution. Bilinear interpolation and transposed convolution are commonly used for upsampling. In bilinear interpolation, the coordinate values of the points to be interpolated are linearly interpolated in X- and Y-axes to restore the image resolution. The transposed convolution is the reverse process of convolution. It decodes the features extracted by convolution to restore the image resolution. In this paper, a transposed convolution layer with a stride of 2 and a convolution kernel of size 2 × 2 is used for upsampling. The image size is doubled after each upsampling.

#### 2.2.4. Activation Function

An activation function is an important part of the CNN model, which is used to increase the nonlinear expression capacity of the CNN model. In this paper, *PReLU* [46], an improved version of *ReLU* [48], is used as the activation function of the new model, which can adaptively learn the parameters from the data. The *PReLU* has the characteristics of fast convergence and low error rate. The calculation formula is as follows:(5)$$PReLU\left(x_{i}\right)=\left\{\begin{array}{cc} x_{i} & x_{i}>0\\ a_{i}x_{i} & x_{i}\leq 0 \end{array}\right.$$
where $a_{i}$ represents the parameter of a learnable rectified unit and i stands for different channels.

### 2.3. Loss Function

The function of weighted cross-entropy loss is used in this paper to reduce the impact of class imbalance in the hyperspectral dataset on the accuracy of the model. First, the overall sample size is divided by the sample size of a single class to obtain the reciprocal of the proportion of the sample size of a single class in the overall sample size. Then, the logarithm of the result of the previous step is obtained with 10 as the base and considered as the weight of each land cover class. Finally, the final loss function using the cross-entropy is obtained as:(6)$$Loss=-\overset{K}{\underset{k=1}{\sum }}\log(\frac{T}{t_{k}})y_{k}\log(p_{k})$$
where K is the number of classes, y and p are the real and the predicted values, respectively, T is the overall sample size, and t is the sample size of a single category.

## 3. Experimental Results and Analysis

### 3.1. Parameter Setting of the Network

All experiments in this paper are completed under the framework of Keras open-source deep learning. The experimental hardware is configured as NVIDIA GeForce RTX 2080Ti GPU with a video memory of 11 GB. Before training, the training data is enhanced by rotating and flipping, and the network weight parameters are initialized by a He-normal distribution initializer [46]. The network training is carried out based on the function of weighted cross-entropy loss and Adam optimizer [49]. The batch size, the initial learning rate and the weight attenuation rate are set as 256, 0.001 and 0.00001, respectively. When the loss of the validation set does not decrease after 10 iterations, the learning rate is adjusted to half of the initial value until the loss of the validation set tends to be stable to end the training.

### 3.2. Evaluation Metrics

Five metrics, namely, Kappa coefficient (Kappa), Mean Intersection over Union (mIoU), weighted average precision (WAP), weighted average recall (WAR) and weighted average F1-score (WAF) are used to validate the performance of the proposed model. Kappa is used to measure the consistency between the predicted and the real values of the multiple classification models. The mIoU is the standard measure in the field of semantic segmentation. These two common metrics will not be listed here. The *WAP*, *WAR* and *WAF* are used to measure the performance of multiple classification models with serious class imbalance. The calculation formulas are:(7)$$WAP=\overset{K}{\underset{k=1}{\sum }}\left(\frac{t_{k}}{T}\times \frac{TP_{k}}{TP_{k}+FP_{k}}\right)$$
(8)$$WAR=\overset{K}{\underset{k=1}{\sum }}\left(\frac{t_{k}}{T}\times \frac{TP_{k}}{TP_{k}+FN_{k}}\right)$$
(9)$$WAF=\frac{2\times WAP\times WAR}{WAP+WAR}$$
where K is the number of classes, T is the overall sample size, $t_{k}$ is the sample size of the class k, while $TP_{k}$, $FP_{k}$, and $FN_{k}$ are the true positive, the false positive, and the false negative of the class k, respectively.

### 3.3. Dataset Preprocessing

#### 3.3.1. Salinas Dataset Partitioning

In this paper, the Salinas public dataset published on the National Aeronautics and Space Administration (NASA) website is used. The Salinas dataset is commonly used in the classification of HSRSI. The dataset is photographed by an Airborne Visible/Infrared Imaging Spectrometer (AVIRIS), with a total of 224 continuous spectral bands, excluding 20 bands absorbed by water. The wavelength range is from 400 nm to 2500 nm, the spatial resolution is 3.7 m, and the image size is 512 × 217. Sixteen land cover classes have been labeled in the Salinas dataset. Together with the background that has not been labeled, 17 classes have been labelled in total, as shown in Figure 5. The Salinas dataset is partitioned by the dataset partition method in this paper, and a total of 112 patches of 32 × 32 are obtained. During the experiment, 66 patches were used for training, 23 were used for validation, and the remaining 23 were used for testing. The sample size of each class in each set is listed in Table 1.

#### 3.3.2. Selecting the Dimension of the Salinas Dataset

In order to select the appropriate dimension, it is necessary to study the influence of the hyperspectral Hughes phenomenon on the accuracy of the model in semantic segmentation. In this paper, the CVCR is used as the dimension reduction standard of the PCA. The CVCRs of 99%, 99.9%, 99.99%, 99.999%, and 100% are adopted to reduce the dimension of the Salinas dataset and five groups of data are obtained. The five data groups are used as the input data of the new algorithm proposed in this paper to compare the segmentation performance of different models. The experimental results show that (Table 2), with the increase of dimension, the accuracy evaluation metrics of the five segmentation approaches all show a trend of first increasing and then decreasing. The optimal evaluation metrics are obtained when the dimension decreases to 31. When the CVCR is low, the dimensionality reduction loses too much information, resulting in poor segmentation performance. When the CVCR is high, the model learns too many nonlinear features from a small number of samples in the training set, and the “dimension disaster” causes the overfitting phenomenon, which affects the performance of the classifier. Compared with 31D, which shows the best performance, the five accuracy evaluation metrics of 3D and 204D are quite different. The results demonstrate that selecting the appropriate dimension can effectively reduce the impact of the Hughes phenomenon on the accuracy of land cover classification of HSRSI.

Figure 6 shows the visualized segmentation results with different CVCRs. It can be seen that in the segmentation map with a CVCR of 99%, the objects that have similar features, including Broccoli_green_weeds_1 and Broccoli_green_weeds_2, Fallow and Fallow_smooth, Grapes_untrained, and Vineyard_untrained, are seriously misclassified (marked with red circles in Figure 6); when the CVCR increases to 99.9% and 99.99%, the misclassification phenomenon decreases. However, when the CVCR reaches 99.999%, the obvious misclassification of Grapes_untrained and Vineyard_untrained appears again. The segmentation results without dimensionality reduction also show that the objects with similar features are misclassified (red circles shown in Figure 6). In general, misclassification mainly occurs among the classes with similar features. With the increase of dimension, misclassification shows a trend of first decreasing and then increasing.

It can be observed from the visualized maps of segmentation performance and the segmentation results of the model with different CVCRs that the PSE-UNet model segmentation results are the best when the CVCR is 99.99% for dimensionality reduction of the Salinas dataset. Therefore, 31D is selected for dimensionality reduction with PCA in subsequent experiments of the Salinas dataset.

### 3.4. Analysis of Experimental Results

#### 3.4.1. Comparative Analysis of Experimental Results of Different Models

1.Comparison of experimental results of different semantic segmentation models

Taking the Salinas dataset as the basic data source, the PSE-UNet network proposed in this paper is compared with the FCN-8S [10], SegNet [11], UNet [12], 3D-UNet [38], and SS3FCN [35]. Among them, FCN-8S, SegNet, and UNet are three classical semantic segmentation networks, 3D-UNet performs well in semantic segmentation of medical hyperspectral images, and SS3FCN is an advanced method for HSRSI sematic segmentation. In order to ensure the objectivity of the experimental results of different models, the training adopted the same network parameter setting, the input data were patches of 32 × 32, and the CVCR was set to 99.99% (31D) for dimensionality reduction with PCA. Each model was tested five times independently, and the final results were the average of the five experimental results. The above-described five evaluation metrics were used to evaluate the accuracy of the experimental results. As shown in Table 3, compared with those of the other four segmentation models, the five metrics of the model proposed in this paper show the best accuracy, and the Kappa coefficient, WAP, WAR, WAF, and mIoU are 93.359%, 95.348%, 95.218%, 95.238%, and 88.508%, respectively. Compared with the suboptimal 3D-UNet algorithm, the Kappa coefficient, WAP, WAR, WAF, and mIoU of the new algorithm in this paper are increased by 1.943%, 1.352%, 1.402%, 1.434%, and 2.846%, respectively. In addition, in the networks of UNet, 3D-UNet, and PSE-UNet that also use the structure of “encoder-decoder”, the number of parameters of the 3D-UNet network using 3D convolution is three times that of the UNet network, while the number of parameters of the network in this paper is only 4.5 M, less than two-thirds of that of the UNet network. Among the six algorithms, the algorithm proposed in this paper has achieved the best accuracy and outstanding segmentation performance.

2.Analysis of land cover classification results

In order to comprehensively analyze the recognition accuracy of the proposed algorithm for different land features, the segmentation effects of FCN-8S, SegNet, UNet, 3D-UNet, SS3FCN, and the proposed algorithm on 17 land cover classes are compared. The experimental results are shown in Table 4. The new algorithm put forward in this paper has the highest F1-score value in the recognition of 11 classes, slightly inferior to 3D-UNet in the classification accuracy of 3 classes, but the difference is no more than 1%, and a lower classification accuracy than SS3FCN on 3 classes. For easily distinguishable land features, such as Soil_vineyard_develop and Corn_senesced_green_weeds, the segmentation accuracy is slightly improved. However, the segmentation accuracy is improved for the easily misclassified classes, such as Lettuce_romaine in different periods. For the classes of Lettuce_romaine_5wk and Lettuce_romaine_6wk, the segmentation accuracy has been increased by 6.424% and 3.812%, respectively, compared with the suboptimal algorithm. The results show that the C-SE module used in the proposed algorithm integrates spatial and dimensional features more effectively, and can extract more discriminative features.

In addition, the confusion matrix in Figure 7 shows that the algorithm proposed in this paper has good segmentation performance for most categories and less misclassification between different classes. For some patches affected by the interaction between the background and land features, there is relatively more misclassification, and the classification accuracy is slightly reduced. For example, the features of Fallow are similar to the features not labeled in the background, and the contour of Lettuce_romaine is a long strip and there are small sample sizes. Therefore, there is relatively more misclassification between these two classes and the background.

3.Comparison of visualized semantic segmentation results with different models

Figure 8 shows the visualized semantic segmentation results with different algorithms. Compared with the other five algorithms, the results of the PSE-UNet algorithm show a clear contour of land cover features, less misclassification, and the best segmentation performance. The FCN-8S algorithm does not adopt the encoder-decoder structure, and the contour and texture of land cover objects are not as clear as those of the other four algorithms. The SegNet algorithm lacks skip connection to integrate deep features, resulting in the loss of detailed features after downsampling several times and a poor visual effect of segmentation. Compared with the FCN-8S and the SegNet, the SS3FCN algorithm generates a more accurate contour of segmentation maps but with more salt and pepper noise, while the UNet algorithm that uses the symmetrical structure of “encoder-decoder” for segmentation achieves excellent performance. Further comparing the experimental results of the three algorithms with the same structure, the UNet algorithm has poor performance on extracting the contour gap of Lettuce_romaine in different periods (marked with the ellipse in Figure 8). Misclassification can easily occur at the intersection of fine boundaries with the 3D-UNet network (marked with white circles in Figure 8). The PSE-UNet algorithm has less misclassification and the contour of land cover features and the real labels match well, mainly because the PSE-UNet adopts the C-SE module that can extract more discriminative features and lead to better segmentation results.

#### 3.4.2. Analysis of Factors Affecting the Performance of the PSE-UNet Model

From the above semantic segmentation experiments of different models, the proposed PSE-UNet model shows the best segmentation results. To further evaluate the performance of the PSE-UNet model, the effects of downsampling times, different downsampling and upsampling methods, and different activation functions on the final segmentation performance of the PSE-UNet are discussed.

1.Effects of downsampling times on model performance

In semantic segmentation, an overly small receptive field may lead to the loss of global information, and an overly large receptive field may degrade the segmentation performance of the model for small targets. Therefore, it is necessary to select appropriate downsampling times to obtain better segmentation accuracy. Different downsampling schemes of 0, 1, 2, 3, and 4 times are set in the experiment to obtain different segmentation accuracies of the PSE-UNet model. As shown in Table 5, the performance of the model increases first and then decreases with the increase of downsampling times. The optimal segmentation accuracy is obtained with two times downsampling.

2.Effects of different downsampling and upsampling methods on the performance of the model

To study the effect of using different downsampling and upsampling methods on the performance of the model, the segmentation performance of the model using max pooling and convolution as the downsampling methods, and bilinear interpolation and transposed convolution as the upsampling methods, is compared and analyzed. The experimental results are shown in Table 6. It can be seen that due to the learning ability of convolution operation during downsampling, better segmentation performance can be obtained using the convolution layer instead of the pooling layer. The model using transposed convolution for upsampling learns more nonlinear features than the model using bilinear interpolation, which improves the accuracy metrics. The model using convolution and transposed convolution for downsampling and upsampling, respectively, has better segmentation performance than the model using the other three methods.

3.Effects of different activation functions on model performance

To verify the impact of different activation functions on the model performance, PReLU [46] and ReLU [48] are selected as the activation functions for experiments to compare and analyze their impact on the segmentation performance of the model. The results in Table 7 show when each segmentation accuracy matrix of the model is higher when using PReLU as the activation function than that of ReLU as the activation function. By adding a small number of parameters, the PReLU function has overcome the problem that the gradient is 0 when the input of the ReLU function is negative and has improved the performance of the model.

## 4. Conclusions

Currently, finding efficient and intelligent methods for the classification of HSRSI is one of the research focuses in remote sensing. The research on semantic segmentation of HSRSI is not deep enough and there is still much room for improvement in the algorithm structure. Therefore, considering the successful application of the UNet algorithm in the classification of 3D medical images, this paper improves the dataset partitioning strategy in the classification of HSRSI based on the non-overlapping sliding window strategy. This paper introduces the CVCR as the standard for PCA dimensionality reduction and discusses how classification accuracy of HSRSI changes with different dimensions. The symmetrical structure of “encoder-decoder” is introduced into the classification of the HSRSI, based on which a new semantic segmentation algorithm PSE-UNet is proposed for classification. In addition, the effects of downsampling times, different downsampling and upsampling methods, and different activation functions on the performance of the proposed PSE-UNet model are discussed. Experiments are carried out based on the Salinas dataset, and the results show that:Based on the non-overlapping sliding window strategy, the judgment mechanism is introduced to improve the patch allocation scheme, which can overcome the disadvantage that not all classes can be included in the training set after the patches are randomly allocated, effectively avoiding information leakage;When selecting different cumulative contribution rates for dimensionality reduction with PCA, the segmentation accuracy shows a trend of first increasing and then decreasing with the increase of the dimension of the dataset used in the experiments. The segmentation results are the best when the CVCR is 99.99%, indicating that choosing the appropriate dimension can effectively weaken the influence of Hughes phenomenon on the classification accuracy of HSRSI;The segmentation performance of the PSE-UNet algorithm is better than the other four popular semantic segmentation algorithms, showing better segmentation accuracy and visualization effect, and less misclassification of land cover classes. Two times downsampling, convolution and transposed convolution for downsampling and upsampling, respectively, and PReLU as the activation function can effectively improve the segmentation accuracy of the PSE-UNet algorithm in semantic segmentation of the Salinas dataset.

In the semantic segmentation experiments with the Salinas dataset, the approach proposed in this paper shows excellent segmentation performance and can be applied to other semantic segmentation tasks of HSRSI. Different from some existing studies, the dataset partitioning strategy used in this paper retains the background pixels, which is more in line with the actual application scenarios. The comprehensive study of the Hughes phenomenon in this paper can provide a reference for the determination of the dimension of the dataset. The proposed PSE-UNet model considers the characteristics of small sample sizes and multiple dimensions of the HSRSI. The symmetrical structure of “encoder-decoder” and the channel attention mechanism adopted in the proposed model have significant application potential in the semantic segmentation of HSRSI. However, the proposed model still has some problems which need to be further studied in the future, such as low segmentation accuracy of low-frequency land cover features, parameter redundancy, and unvalidated generalization ability.

## Figures and Tables

**Figure 1 sensors-22-09678-f001:**
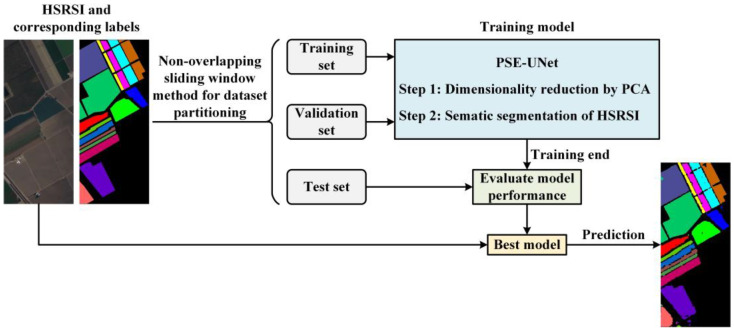
Flowchart of the proposed procedure.

**Figure 2 sensors-22-09678-f002:**
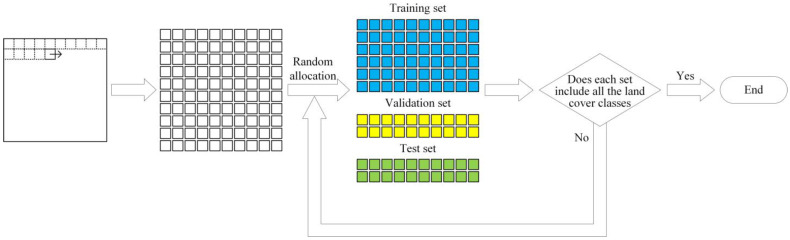
Dataset partition method.

**Figure 3 sensors-22-09678-f003:**
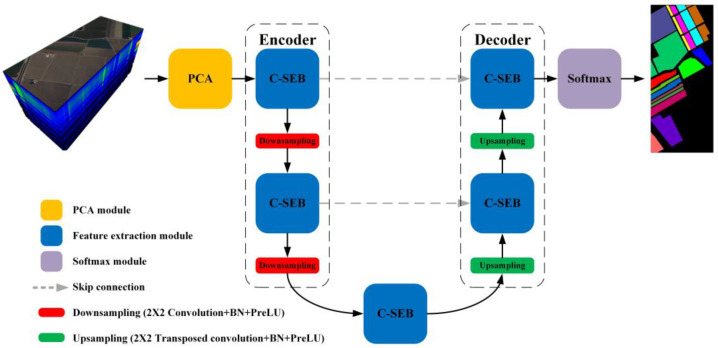
Structure of the proposed PSE-UNet Model.

**Figure 4 sensors-22-09678-f004:**
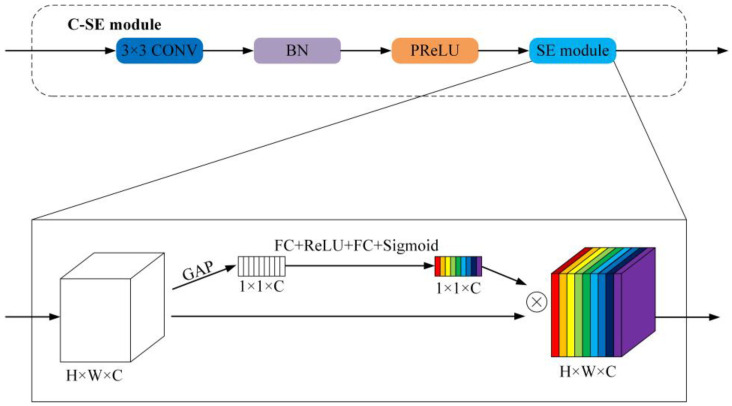
Structure of the C-SE module.

**Figure 5 sensors-22-09678-f005:**
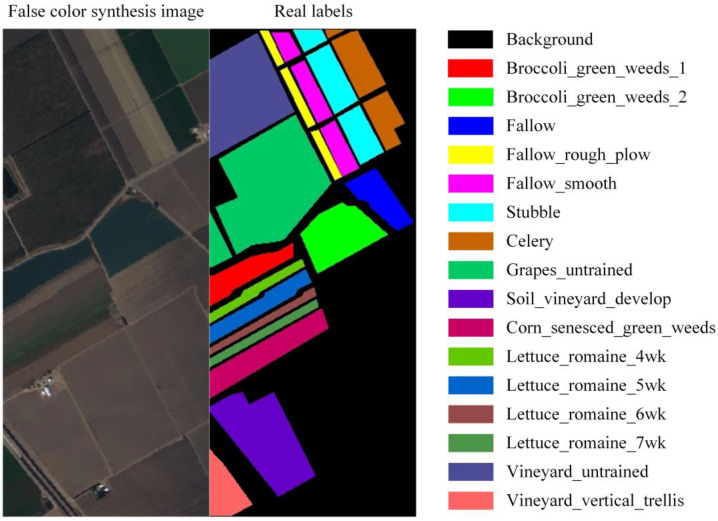
Salinas dataset.

**Figure 6 sensors-22-09678-f006:**
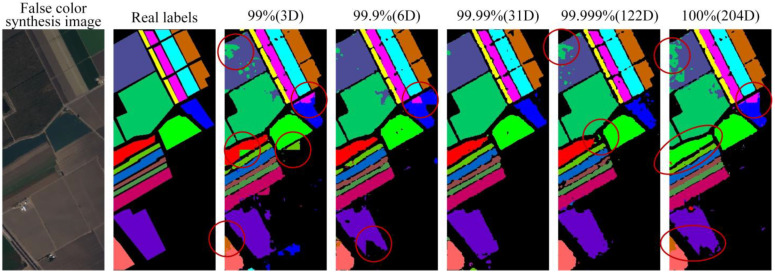
Comparison of visualized segmentation results with different CVCRs.

**Figure 7 sensors-22-09678-f007:**
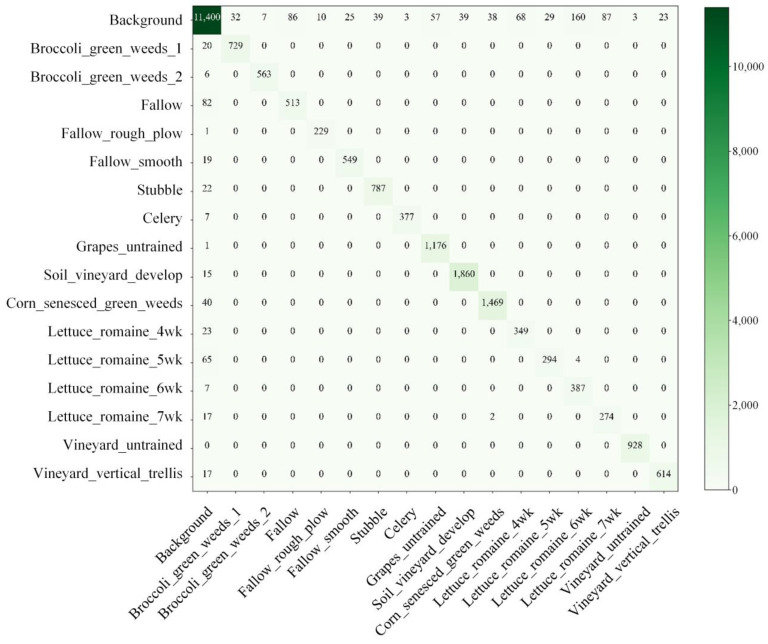
Confusion matrix of classification results of PSE-UNet model.

**Figure 8 sensors-22-09678-f008:**
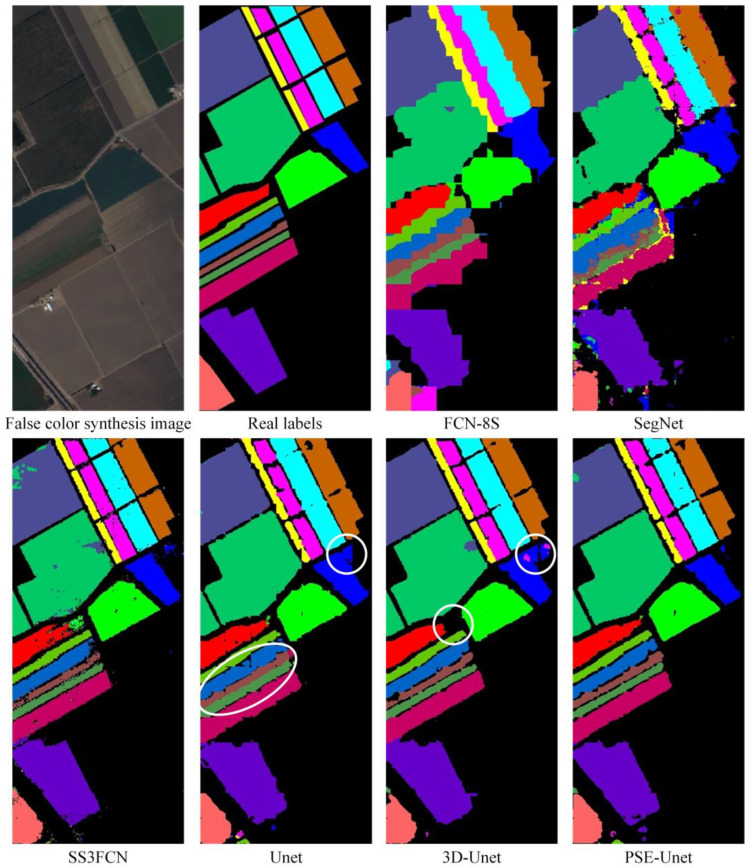
Visualized segmentation results with different algorithms.

**Table 1 sensors-22-09678-t001:** Sample size of each class in each set after partitioning the Salinas dataset.

Class	Train	Val	Test	Total
Background	38,570	9883	12,106	60,559
Broccoli_green_weeds_1	832	428	749	2009
Broccoli_green_weeds_2	1935	1222	569	3726
Fallow	1214	167	595	1976
Fallow_rough_plow	1011	153	230	1394
Fallow_smooth	1659	451	568	2678
Stubble	2641	509	809	3959
Celery	2457	738	384	3579
Grapes_untrained	6368	3726	1177	11,271
Soil_vineyard_develop	1882	2446	1875	6203
Corn_senesced_green_weeds	1303	466	1509	3278
Lettuce_romaine_4wk	290	406	372	1068
Lettuce_romaine_5wk	1090	474	363	1927
Lettuce_romaine_6wk	285	237	394	916
Lettuce_romaine_7wk	476	301	293	1070
Vineyard_untrained	4676	1664	928	7268
Vineyard_vertical_trellis	895	281	631	1807

**Table 2 sensors-22-09678-t002:** Comparison of segmentation performance of the model with different CVCRs.

CVCR (Dimension)	Kappa (%)	WAP (%)	WAR (%)	WAF (%)	mIoU (%)
99% (Dimension 3)	82.195 ± 3.091	88.742 ± 1.412	86.816 ± 2.057	86.966 ± 2.018	70.993 ± 1.297
99.9% (Dimension 6)	89.453 ± 1.879	93.126 ± 0.462	92.276 ± 1.142	92.448 ± 1.026	81.830 ± 6.092
99.99% (Dimension 31)	93.359 ± 0.197	95.348 ± 0.143	95.218 ± 0.091	95.238 ± 0.104	88.508 ± 0.473
99.999% (Dimension 122)	90.282 ± 1.616	93.306 ± 0.578	93.024 ± 0.841	92.888 ± 1.038	83.023 ± 4.089
100% (Dimension 204)	83.739 ± 0.063	87.976 ± 0.119	88.148 ± 0.029	87.424 ± 0.056	69.932 ± 0.619

**Table 3 sensors-22-09678-t003:** Comparison of accuracy of different algorithms.

Algorithm	Kappa (%)	WAP (%)	WAR (%)	WAF (%)	mIoU (%)	Number of Parameters
FCN-8S	77.428 ± 1.779	87.468 ± 1.580	82.256 ± 1.077	82.684 ± 1.121	67.840 ± 3.271	128.2 M
SegNet	76.807 ± 3.850	85.222 ± 6.578	82.822 ± 1.940	83.040 ± 2.657	62.315 ± 8.942	6.7 M
UNet	89.602 ± 1.855	93.326 ± 0.642	92.316 ± 1.124	92.480 ± 1.066	83.414 ± 3.644	8.3 M
3D-UNet	91.416 ± 0.120	93.996 ± 0.125	93.816 ± 0.055	93.804 ± 0.080	85.662 ± 0.707	22.5 M
SS3FCN	89.981 ± 1.783	93.284 ± 0.402	92.743 ± 1.016	92.734 ± 1.059	83.246 ± 4.033	3.7 M
PSE-UNet	93.359 ± 0.197	95.348 ± 0.143	95.218 ± 0.091	95.238 ± 0.104	88.508 ± 0.473	4.5 M

**Table 4 sensors-22-09678-t004:** Comparison of segmentation accuracy for different classes with different algorithms (F1-score %).

Class	FCN-8S	SegNet	UNet	3D-UNet	SS3FCN	PSE-UNet
Background	81.806	86.196	92.350	94.036	94.357	95.362
Broccoli_green_weeds_1	83.412	85.044	94.888	95.610	90.247	95.774
Broccoli_green_weeds_2	79.636	92.552	96.480	98.578	98.310	98.730
Fallow	62.554	36.370	78.816	74.994	88.400	83.568
Fallow_rough_plow	85.368	63.460	93.276	96.166	97.254	96.570
Fallow_smooth	89.614	80.654	92.416	94.980	94.898	95.438
Stubble	89.244	87.100	96.048	96.806	95.992	96.342
Celery	93.128	92.674	97.518	98.466	97.904	98.568
Grapes_untrained	96.764	88.196	98.196	98.616	86.938	97.790
Soil_vineyard_develop	95.226	93.088	96.622	96.324	95.144	98.034
Corn_senesced_green_weeds	80.834	85.634	92.054	95.610	95.633	96.320
Lettuce_romaine_4wk	68.670	56.990	86.704	89.004	90.315	88.436
Lettuce_romaine_5wk	50.878	46.132	71.188	82.596	79.451	89.020
Lettuce_romaine_6wk	58.980	29.998	76.200	70.302	75.928	80.012
Lettuce_romaine_7wk	63.562	41.038	78.326	86.028	83.697	86.662
Vineyard_untrained	92.362	89.992	98.784	99.894	79.650	99.728
Vineyard_vertical_trellis	75.280	87.122	94.706	89.878	92.423	94.814

**Table 5 sensors-22-09678-t005:** Comparison of performance with different downsampling times.

Downsampling Times	Kappa (%)	WAP (%)	WAR (%)	WAF (%)	mIoU (%)
0	92.339 ± 0.291	94.932 ± 0.100	94.384 ± 0.164	94.480 ± 0.149	87.811 ± 0.537
1	93.222 ± 0.096	95.364 ± 0.038	95.068 ± 0.057	95.116 ± 0.060	88.443 ± 0.456
2	93.359 ± 0.197	95.348 ± 0.143	95.218 ± 0.091	95.238 ± 0.104	88.508 ± 0.473
3	91.526 ± 0.312	94.160 ± 0.201	93.858 ± 0.152	93.918 ± 0.163	86.037 ± 0.872
4	86.163 ± 2.114	90.552 ± 1.323	89.882 ± 1.198	89.870 ± 1.399	78.249 ± 6.961

**Table 6 sensors-22-09678-t006:** Comparison of segmentation performance of the model with different downsampling and upsampling methods.

Downsampling and Upsampling Methods	Kappa (%)	WAP (%)	WAR (%)	WAF (%)	mIoU (%)
Max pooling + Bilinear interpolation	92.347 ± 0.172	94.816 ± 0.044	94.432 ± 0.101	94.510 ± 0.086	87.462 ± 0.506
Convolution + Bilinear interpolation	93.090 ± 0.081	95.318 ± 0.055	94.980 ± 0.037	95.040 ± 0.045	88.365 ± 0.217
Max pooling + Transposed convolution	92.507 ± 0.227	94.946 ± 0.042	94.566 ± 0.138	94.650 ± 0.113	87.587 ± 0.394
Convolution + Transposed convolution	93.359 ± 0.197	95.348 ± 0.143	95.218 ± 0.091	95.238 ± 0.104	88.508 ± 0.473

**Table 7 sensors-22-09678-t007:** Comparison of segmentation performance of the model with different activation functions.

Activation Function	Kappa (%)	WAP (%)	WAR (%)	WAF (%)	mIoU (%)
ReLU	92.841 ± 1.286	95.088 ± 0.402	94.818 ± 0.714	94.870 ± 0.627	87.635 ± 2.199
PReLU	93.359 ± 0.197	95.348 ± 0.143	95.218 ± 0.091	95.238 ± 0.104	88.508 ± 0.473

## Data Availability

The Salinas dataset utilized in this study are freely available at http://www.ehu.eus/ccwintco/index.php/Hyperspectral_Remote_Sensing_Scenes (accessed on 25 October 2022).

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
