# Peer review of "Semantic Segmentation of Hyperspectral Remote Sensing Images Based on PSE-UNet Model"

_sensors, 2022, doi:10.3390/s22249678_

Round 1
Reviewer 1 Report
Summary: The manuscript presents a deep learning based approach for semantic segmentation of hyperspectral remote sensing images using a PSE-UNet. A dataset partitioning scheme is presented where each image is divided into non-overlapping patches and care is taken so that the test and training datasets contain all the classes. The dimensions of the hyperspectral images are reduced using PCA with cumulative variance contribution rate (CVCR) as the metric. A PSE-UNet is trained on the reduced dimension images for predicting semantic labels. Extensive analysis of the performance for the number of dimensions, the choice of the activation function, the downsampling and upsampling approach and the choice of the deep network architecture is presented.
Comments:
1. The manuscript is well written and easy to follow.
2. There are adequate citations.
3. One of the key contributions that you state in your work is the dataset partitioning scheme using a non overlapping sliding window and a class balancing approach. This is a well known technique in computer vision. Therefore, it is not entirely novel and must not be part of the the paper's research contribution. Please reword the document accordingly.
4. Dimensionality reduction with PCA and CVCR is also well known, although its application to hyperspectral data is new. Please reword the document accordingly.
5. The comparative analysis for every step along the pipeline is well presented and provides a strong basis for the choices made in constructing the framework.
6. In table 3, you present the size of the weights file in MB. It it also good to present the number of parameters (weights. biases) of each model if possible as this is a better indication of the size of a network. Along with this, it would also be of use to present the training and inference times of the network.
7. Minor typographic errors are present. E.g. matric -> metric in the abstract
Reviewer 2 Report
This paper presents a semantic segmentation method in hyperspectral remote sensing images. The method is sound. The experiments show the effectiveness of the proposed method. However, there are several concerns about this study. The authors need to revise this paper before it can be accepted.
- The motivation of this study should be further enhanced.
- The challenge of this study needs to be clearly indicated. Besides, what is the relationship between the proposed method and the challenge.
- Please provide a flowchart of the proposed method.
- The difference between the proposed method and the previous studies should be indicated.
- More papers need to be cited for enriching the literature review, e.g. Video salient object detection using dual-stream spatiotemporal attention; Industrial pervasive edge computing-based intelligence IoT for surveillance saliency detection.
- Could the section about PCA be refined, because the introduction of PCA is seemly the background knowledge.
- What is the novelty of the PSE-UNet. It is hard to find it in the method section.
- It seems that the proposed method is not compared with the hyperspectral remote sensing Image segmentation method. Please explain the reason.
- The image resolution should be improved.
- Some grammatical errors.
Round 2
Reviewer 2 Report
The authors have addressed the comments raised by the reviewers. I have no further question.